# Dietary Patterns in Pregnancy and the Postpartum Period and the Relationship with Maternal Weight up to One Year after Pregnancy Complicated by Gestational Diabetes

**DOI:** 10.3390/nu15194258

**Published:** 2023-10-05

**Authors:** Letícia Machado Dias, Maria Inês Schmidt, Álvaro Vigo, Michele Drehmer

**Affiliations:** 1Postgraduate Studies Program in Epidemiology, Department of Social Medicine, School of Medicine, Federal University of Rio Grande do Sul, 2400 Ramiro Barcelos St., 2nd Floor, Porto Alegre 90035-003, Brazil; leticiadiasnutricionista@gmail.com (L.M.D.); maria.schmidt@ufrgs.br (M.I.S.); alvaro.vigo@ufrgs.br (Á.V.); 2Hospital de Clínicas de Porto Alegre, Porto Alegre 90035-903, Brazil; 3Postgraduate Studies Program in Food, Nutrition and Health, Department of Nutrition, School of Medicine, Federal University of Rio Grande do Sul, Porto Alegre 90035-003, Brazil

**Keywords:** dietary pattern, gestational diabetes, postpartum weight, cohort study

## Abstract

This multicentric cohort study aimed to describe changes in dietary patterns during pregnancy and postpartum and the association with BMI variation at six and twelve months postpartum in women with gestational diabetes mellitus (GDM). Between 2014 and 2018, we enrolled women with GDM in prenatal clinics of the Brazilian National Health System and followed them for one year postpartum. The dietary patterns during pregnancy and the postpartum period were obtained by factorial analysis. The relationship between these patterns and variation in postpartum BMI was evaluated by Poisson regression with robust variance adjusted for confounders. We identified three dietary patterns in 584 women, two healthy (generally healthy and Dash type), which were associated with less weight gain (RR 0.77 CI 95% 0.62–0.96 and RR 0.71 CI 95% 0.57–0.88, respectively). The high-risk pattern (based on ultra-processed, high-calorie foods and sweetened drinks) was associated with weight gain (RR 1.31 CI 95% 1.07–1.61 and RR 1.26 CI 95% 1.01–1.59) in six and twelve months postpartum, respectively. Although the participants learned about healthy dieting during pregnancy, dietary habits worsened from pregnancy to postpartum, especially, with lower consumption of fruits and dairy and higher consumption of sweetened beverages, with consequent weight gain postpartum. Postpartum support is needed to prevent weight gain and obesity.

## 1. Introduction

Obesity is a growing problem worldwide that caused 2.4 million deaths and 70.7 million disability-adjusted life years in females in 2017 [1]. A high body mass index (BMI) impacts reproductive health, affecting the mother and child, with intergenerational consequences [2,3]. The trajectory of maternal weight from the beginning of pregnancy through the postpartum period is associated with a risk of obesity in childhood [4], sharing genetic predispositions, intrauterine effects (particularly of increased insulin concentrations on metabolic regulation), and environmental factors such as diet quality, feeding behaviors, and obesogenic home and community environments [5]. In Brazil, gestational weight gain has been shown to have an effect on child BMI across three generations [6].

Among women of reproductive age living in Brazilian cities, the prevalence of overweight and obesity is 49% (46–52.1) and 17.9% (15.5–20.3) in the age group of 25 to 34 years; in those aged between 35 and 44 years, 56.3% (53.7–58.9) and 21.1% (18.9–23.4), respectively [7]. A high pre-gestational BMI is associated with excessive gestational weight gain, the development of gestational diabetes mellitus (GDM), and postpartum weight retention [8,9,10,11]. In addition, women with GDM have a higher future risk of type 2 diabetes mellitus [12]. 

The pattern of food consumption is associated with maternal obesity and hyperglycemia during pregnancy, which can lead to complications in women and may affect the child’s susceptibility to diseases in adulthood [13]. In a Brazilian study, it was demonstrated that the consumption of ultra-processed foods (UPF) was associated with obesity in women with GDM and the consumption of fresh or minimally processed foods had an inverse relationship with obesity [14]. In women participating in the Nurses’ Health Study II with a history of GDM after being followed for 20 years, it was found that the highest adherence to a healthy eating score (aHEI-2010, Mediterranean diet and DASH diet) was significantly associated with less weight gained, regardless of other factors such as baseline BMI, changes in physical activity and smoking [15].

Systematic reviews show an inverse relationship between greater adherence to “Vegetable”, “Mediterranean-style”, and “Prudent” dietary patterns, characterized by a high intake of vegetables, fruits, whole grains, nuts, legumes, fish, poultry, and low-fat dairy, which are part of the “in nature and minimally processed” food group, and the risk of GDM [16,17]. Pre-gestational ultra-processed food consumption was associated with an increased risk of GDM in Spanish women aged 30 years or more but not in younger women [18]; however, to the best of our knowledge, evaluations of dietary patterns in the recent postpartum period and weight in women with this condition are scarce. Given the importance of assessing food consumption as a modifiable factor, related to the variation in weight and women’s health, and the scarcity of data on changes in dietary patterns during pregnancy and postpartum in women with GDM, the present study aims to describe the changes in food consumption between the gestational and postpartum periods of women with GDM and its association with BMI variation at six months and one year postpartum.

## 2. Materials and Methods

### 2.1. Design and Sample

We conducted a prospective cohort study of women with GDM recruited during pregnancy and followed until one year postpartum. The recruitment cohort was planned with the goal of identifying eligible women for a randomized clinical trial for diabetes prevention, LINDA-Brasil [19]. The study was conducted in accordance with the guidelines of the Declaration of Helsinki and all subjects gave their informed and signed consent for inclusion before they participated in the study. The protocol was approved by the Ethics Committee of the Hospital de Clínicas de Porto Alegre (Project 120097, 4 May 2012).

### 2.2. Recruitment

We recruited women from referral centers in the Brazilian National Health System in two southern cities (Porto Alegre and Pelotas) and one northeastern city (Fortaleza) between March 2014 and January 2018. Face-to-face interviews included questions regarding socioeconomic, clinical, nutritional, and food consumption characteristics. In addition, we obtained pregnancy information from the women’s prenatal cards.

### 2.3. Postpartum Follow-Up

About one month after recruitment, we started follow-up telephone calls, the first one during the last few weeks of pregnancy, the second within the first month after delivery, and the other ones at 2, 6, and 12 months after delivery. We assessed women’s weight, the date of measurement, and breastfeeding. At six months postpartum, we also asked about their usual food consumption. A trained and certified team conducted interviews. 

### 2.4. Food Consumption

Food consumption variables were collected using a brief food frequency questionnaire (short-FFQ) focused on risk and protection food markers adapted from VIGITEL and SISVAN [20,21]. The short-FFQ, applied during pregnancy and six months postpartum, was composed of 16 food items or groups of food, namely: raw salads; cooked salads; fruits; natural juices; beans; semi-skimmed and skimmed milk and derivatives (yogurt); whole milk and derivatives; red meats with visible fats; red meats without visible fats; chicken meat; fish; fried foods (chips, packaged potatoes, fried snacks); salty snacks (savory cookies, packaged snacks); cookies and sweets (sweet cookies, stuffed cookies, candies, chocolates, chocolates); soft drinks or juices in powder; and sausages (sausage, salami, sausage, ham, turkey breast).

During pregnancy, women were asked about their food frequency consumption since they became pregnant. In the postpartum period, women were asked about their usual consumption since the baby was born.

### 2.5. Outcome 

The outcome was the variation in BMI between two months and six months postpartum and between two months and twelve months postpartum. The participants mentioned their most recent weight in each of the calls. Before the follow-up call, a text message was sent to the participant’s cell phone, asking them to measure their weight before the scheduled phone contact. We obtained patient height information from prenatal cards or via self-report. 

We calculated the change in body mass index (BMI) at postpartum from 2 months to 6 and 12 months postpartum. If the BMI increased between 2 and 6 months postpartum, we considered it a risk for weight gain (BMI at six months higher than BMI at two months postpartum). The same calculation was used for the difference between BMI at 2 months and 12 months postpartum (BMI at 12 months higher than BMI at 2 months postpartum was considered a risk for weight gain).

### 2.6. Covariates

Covariate information included age, monthly family income (minimum wage: USD 284), marital status, number of children, self-reported skin color, smoking before and during pregnancy, and pre-gestational weight. Pre-pregnancy BMI was calculated as self-reported pre-gestational weight (kg)/height (m^2^), and total gestational weight gain was classified according to the Institute of Medicine 2009 [22]. We evaluated if women were breastfeeding at six and twelve months postpartum. 

### 2.7. Statistical Analysis

We used the absolute and relative frequency to describe the sample and food items consumed. The consumption of food items was grouped into three categories according to the weekly frequency of consumption, 0 to 2 times a week, 3 to 4 times a week, and 5 to 7 times a week.

Dietary patterns were characterized using factor analysis, with initial factors extracted using the principal component method. All 16 food items were standardized for consumption units per week. The Kaiser–Meyer–Olkin test (KMO > 0.6) and Bartlett’s sphericity test (*p* < 0.05) were applied to verify that all assumptions of the analysis were met. The factors were rotated orthogonally by the varimax method, and those with eigenvalues > 1 were retained. To determine the number of factors to be part of the food pattern, we considered the scree plot graph and the interpretability of the factors using food items with loads > 0.30. Three dietary patterns were identified and named according to food items with high loads and later categorized into terciles.

Poisson regression analysis with robust variance was used to estimate the adjusted association between the terciles of dietary patterns and the change in BMI at 6 months and 1 year postpartum (differences in BMI between 2 and 6 months > 0 and BMI between 2 and 12 months > 0 were considered a risk category). 

Confounders were adjusted in 3 models: model 1 included demographic covariates (age, skin color, income, smoking during pregnancy, number of previous pregnancies, and center of investigation); model 2 included the variables in model 1 and the pre-gestational BMI and gestational weight gain; and model 3 included the variables in model 2 and breastfeeding. 

Poisson regression with robust variance was also performed to investigate adjusted associations of food items with a higher factor load in the dietary patterns identified with BMI modification at six and twelve months postpartum. The variables were adjusted using models similar to those previously mentioned. We presented risk ratios followed by the 95% confidence interval (95% CI). This allows the assessment of precision and indicates the lower and upper bounds that may fall within the statistical significance range. All analyses were performed using SPSS version 21 software [23]. 

## 3. Results

As seen in Figure 1, we enrolled 2860 women with 28 weeks or more of pregnancy, 18 years or older diagnosed with GDM, without known diabetes before pregnancy, and without positive HIV serology. We excluded 536 participants with missing data on demographics, pre-pregnancy BMI, and food frequency variables and thus our sample comprised 2324 women. Complete data on food consumption at six months postpartum was available for 1098 participants and at one year for 584 participants.

As seen in Table 1, sociodemographic and clinical characteristics were similar for those in the initial sample (n = 2324) and for the two sets of participants with complete information during follow-up (n = 1098 and n = 584). A few exceptions might be fewer white women and fewer very poor women in the follow-up samples.

Table 2 shows the percentage of food consumption during pregnancy and postpartum in women who answered the food questionnaire during pregnancy (n = 2324) and for those who also answered the questionnaire during the postpartum period (n = 1098). Frequent (five to seven times a week) consumption of vegetables, fruits, and legumes decreased in the postpartum period. We observed an absolute decrease of 9.5% for raw salad, 4.7% for cooked salad, and 26.5% for fruit. Among dairy products, we observed an absolute decrease of 30.3% for skimmed dairy products and 34.4% for whole dairy products. The consumption of red meat and chicken, in general, did not change. As for hypercaloric and/or ultra-processed foods, the frequent consumption of sweetened beverages increased by 9.5% in the postpartum period. The consumption of fried foods, salty snacks, and sweets showed a slight decrease.

As presented in Table 3, we identified three dietary patterns during pregnancy: a “risk pattern” composed mainly of ultra-processed and high-calorie foods; a “healthy pattern”, composed mainly of lean meats, vegetables, and fruits; and a “DASH-type diet pattern”, consisting mainly of low-fat dairy products, vegetables, legumes, and vegetables. In the postpartum period, three patterns were identified: a “risk pattern” similar to that found in pregnancy, consisting mainly of ultra-processed and high-calorie foods; a “mixed pattern”, consisting of natural juice, salty snacks, fruit, and chicken; and a “DASH-type diet” pattern, similar to that found in pregnancy.

Figure 2 and Figure 3 show the adjusted associations between food consumption patterns during pregnancy and changes in BMI during follow-up at six and twelve months postpartum, respectively. Adherence to a dietary risk pattern in pregnancy (third tertile compared to first tertile) was associated with a 31% increase (RR 1.31; 95% CI 1.07–1.61) in maternal BMI between two and six months postpartum. Consumption of a healthy dietary pattern (second tertile compared to the first tertile) was associated with a 23% reduction (RR 0.77 95% CI 0.62–0.96) in maternal BMI between two and six months post-childbirth. The DASH pattern during pregnancy was not associated with postpartum BMI changes and no associations were observed between dietary patterns during pregnancy and BMI changes at 12 months postpartum. 

Figure 4 and Figure 5 represent the adjusted associations between postpartum food consumption patterns and changes in follow-up BMI. In the postpartum period, adherence to the dietary pattern of risk (third tertile compared to first tertile) increased by 26% (RR 1.26 95% CI 1.01–1.59) maternal BMI at six months after delivery. The postpartum mixed dietary pattern was not significantly associated with postpartum changes in maternal BMI. In turn, adherence to the DASH-type diet pattern in the postpartum period was associated with a 31% decrease in maternal BMI at 12 months after delivery (RR 0.71 95% CI 0.57–0.88).

Examining individual food items, after adjustment, we found that the high intake of fried foods during pregnancy (5–7/week) was associated with an 83% (RR 1.83 95% CI 1.43–2.34) maternal BMI increase at six months after delivery and with a 48% (RR 1.48 95% CI 1.06–2.07) maternal BMI increase at one year postpartum (Appendix A). The high consumption of this food item, during postpartum, increased maternal BMI by 87% (RR 1.87 95% CI 1.40–2.50) at six months after delivery and by 52% (RR 1.52 95% CI 1.00–2.29) at one year postpartum (Appendix A). Similarly, the high consumption of sweetened beverages during pregnancy also increased maternal BMI by 31% (RR 1.31 95% CI 1.09–1.58) at six months after delivery and by 29% (RR 1.29 95% CI 1.08–1.55) at twelve months after delivery (Appendix A). A high intake of sweetened beverages during the postpartum period led to an increase in maternal BMI by 27% (RR 1.27 95% CI 1.04–1.55) at 1 year after delivery. Finally, high consumption of cookies and sweets during postpartum led to an increase in maternal BMI by 44% (RR 1.44 95% CI 1.14–1.82) at six months after delivery (Appendix A).

In contrast, participants who had a high consumption of raw salad during pregnancy had a considerably lower maternal BMI at six months postpartum (RR 0.77 95% CI 0.62–0.97) (Appendix A). For cooked salad, we found associations between consumption during pregnancy and maternal BMI at six months and one year after delivery. Participants who had a high consumption of cooked salad during pregnancy had a considerably lower maternal BMI at six and twelve months postpartum (RR for 5–7/week vs. 0–1/week = 0.76 95% CI 0.59–0.97 and RR = 0.75 95% CI 0.59–0.95, respectively) (Appendix A). We found a strong and graded negative association for participants who had a high consumption of raw salad during postpartum and maternal BMI at one year after delivery (RR for 2–4/week vs. 0–1/week = 0.80 CI 95% 0.65–0.97; RR for 5–7/week vs. 0–1/week = 0.68 CI 95% 0.55–0.84) (Appendix A). We also found that, by enhancing fruit consumption during postpartum, participants had considerably lower maternal BMI at one year after delivery (RR for 2–4/week vs. 0–1/week = 0.79 95% CI 0.65–0.96 and RR for 5–7/week vs. 0–1/week = 0.76 95% CI 0.62–0.92). Consumption of chicken during the postpartum period also reduced BMI at six months (RR 0.70 95% 0.53–0.91) (Appendix A).

## 4. Discussion

In this cohort of women with gestational diabetes recruited while pregnant, we found a considerable worsening of their dietary habits postpartum, with an associated weight gain. A switch from less processed foods to more ultra-processed foods was associated with lower consumption of salads, fruits, cooked vegetables, and dairy products, and higher consumption of sweetened beverages. The two healthy patterns identified (a generally healthy and a DASH-type diet) were associated with less weight gain (varying from RR 0.77 CI 95% 0.62–0.96 to RR 0.71 CI 95% 0.57–0.88, respectively). The high-risk pattern during pregnancy and postpartum period was associated with weight gain (varying from RR 1.31 95% CI 1.07–1.61 to RR 1.26 95% CI 1.01–1.59). Specific food items related to an elevated body mass index at postpartum were fried food, sweetened beverages, cookies, and sweets during postpartum; with lower elevations associated with the consumption of salads, fruits, and cooked vegetables. 

Women with GDM receive more intensive care during prenatal care, which includes meal planning and food intake monitoring [24], and concerns about fetal development and health are possibly the main motivators for adherence to habits understood as healthy [11,25]. However, after discharge from the maternity hospital, women return to the primary care units of origin and follow-up care in the puerperium becomes less intense compared to the gestational period [26]. Postpartum weight gain in these high-risk women, often associated with inadequate eating habits and low physical activity, is likely a key prognostic factor and mediator in the development of T2DM after pregnancy with GDM [27].

Worsening dietary patterns in women postpartum who had GDM have been reported previously [28,29]. Barriers to a healthy diet after delivery include cost, lack of time, doubts about which foods to eat, lack of motivation, personal or cultural food preferences, and a lack of knowledge on how to prevent T2DM. Women expressed a sense of abandonment after the intense treatment and management of GDM during pregnancy [26]. 

We found that two healthy patterns, “generally healthy” and “DASH type”, materialized as protectors of postpartum weight gain. Evidence indicates that adherence to healthy dietary patterns during pregnancy, such as a Mediterranean diet, rich in antioxidants and phenolic compounds; the New Nordic Diet, based on the consumption of fruits, root vegetables, cabbages, potatoes, oatmeal, whole grains, wild fish, berries, and milk; and the Dash Diet, which is rich in fruits, vegetables, whole grains, and low-fat dairy products, protects against postpartum weight gain [15,30,31]. In the Misc Cohort Study, a negative association was observed between high adherence to a Mediterranean diet during pregnancy and weight retention at 2 and 6 months postpartum [30]. Women with higher adherence to the New Nordic Diet pattern during pregnancy had, on average, lower post-partum BMI trajectories and slightly less weight gain up to 8 years post-delivery compared with the lower New Nordic Diet adherers [31]. Finally, women with a history of GDM (n = 3397), followed from 1991 to 2011, or until a diagnosis of type 2 diabetes or other chronic diseases, in the highest quintile (Q5) of diet change (most improvement in quality), gained significantly less weight per four-year period than the lowest quintile (Q1; decrease in quality), independent of other risk factors (four-year weight change, DASH: Q5 = 0.64 kg vs. Q1 = 2.75 kg) [15]. Dietary patterns that result in lower postpartum maternal weight have a greater chance of reducing the risk of T2DM in women with a history of GDM [27].

We also found a high-risk dietary pattern based on sweetened drinks may increase the risk of postpartum weight gain. This finding corroborates with other studies. In a recent US cohort study, high ultra-processed food intake during pregnancy was associated with greater gestational weight gain, postpartum weight retention, and pregnancy C-reactive protein [32]. In Brazilian women, greater ultra-processed food intake before and during pregnancy has been associated with higher gestational weight gain and risk of GDM and obesity [14,33]. In the LINDA-Brasil study, we found that the consumption of sweetened beverages during the perinatal period is associated with higher postpartum BMI at 6 and 12 months (RR 1.31 95% CI 1.09–1.58 and RR 1.29 95% CI 1.08–1.55, respectively, Appendix A). A similar finding was observed in a study of 348 mothers in Boston that found an association between greater sugar-sweetened beverage intake frequency in the third trimester and weight retention at 6 months postpartum (OR 1.37 95% CI 1.10–1.75). Avoiding sugar-sweetened beverages may reduce the risk of excess weight retention [34].

An important finding that should be highlighted is the increased consumption of sweetened beverages between the gestational and postpartum periods and a drastic decrease in milk consumption. Consumption of skimmed dairy products between five and seven times a week was 32.3% in pregnancy and dropped to just 2% postpartum and consumption of full-fat dairy varied from 38.4% in pregnancy to only 4% postpartum. It is possible that postpartum dairy consumption is replaced by sweetened beverages, as the consumption of soft drinks and artificial juices five to seven times a week increased from 25.6% in pregnancy to 35.1% postpartum. It is likely that women with GDM were instructed to restrict the consumption of sweetened beverages, returning postpartum to their pre-pregnancy habits. The consumption of sweetened beverages is related to an unfavorable evolution of glycemic levels during pregnancy and is positively associated with the risk of type 2 diabetes in the adult population [35,36].

In addition, the risk pattern identified in pregnancy and postpartum including high-energy-density foods such as fast food, ultra-processed food, and sugar-sweetened beverages was associated with an increased risk of increments in BMI at 6 and 12 months postpartum (ranging between RR 1.26 and 1.31). These food groups are considered proinflammatory and proinsulinemic because they increase the plasma levels of inflammatory biomarkers, such as C-reactive protein and C-peptide, which are proxies for inflammation and insulin, respectively. With the consumption of these foods, it is possible to achieve a high calorie value even from consuming small portions, resulting in an association with substantial long-term weight gain regardless of energy intake, as well as worsening glycemic levels in these high-risk women to the development of T2DM [34,37,38].

Alternatively, the DASH-type pattern we identified includes foods high in fiber and other satiating constituents, such as fruits, vegetables, and dairy, which have been associated with less long-term weight gain among US adults [15]. It is possible that eliminating ultra-processed foods including sugar-sweetened beverages, known contributors to weight gain, would be expected to lead to less long-term weight gain. Other potential intermediates of diet quality independent of its effect on caloric intake with weight change are possible, including inflammation, changes to the gut microbiota, or other unknown mechanisms [15].

The present study has some limitations. The instrument that evaluated food consumption was composed of a small number of food items, but it was based on the VIGITEL [20] and SISVAN [21] questionnaires, important assessment tools that are brief food frequency questionnaires (short-FFQ) focused on risk and protection food markers for non-communicable chronic diseases in the Brazilian population. Participants were asked about food consumption in person during pregnancy and by phone call in the six months postpartum. The non-face-to-face questions can generate difficulties in understanding. All pre- and post-pregnancy weight variables were self-reported. Furthermore, at different stages, the sample showed a slight change in the proportion of sociodemographic (skin color, marital status, family income) and clinical (gestational weight gain) aspects. These factors have an effect on food consumption and therefore could exert a selection bias effect. However, the differences between samples were small regarding the aspects evaluated (Table 1 and Table 2), which suggests that, if present, these views would have little effect.

## 5. Conclusions

In summary, our study contributes to the understanding of associations between food consumption during pregnancy and postpartum and the maternal nutritional status up to one year after pregnancy in women who had a recent diagnosis of GDM. This is the first study to evaluate the relationship between food consumption patterns during pregnancy and six months after pregnancy and changes in BMI at six and twelve months in relation to BMI two months after delivery in women with GDM. Worsening dietary patterns from pregnancy to postpartum result in postpartum maternal weight gain, which has a greater chance of enhancing the risk of T2DM in women with a history of GDM. Further prospective studies are needed to assess, in addition to BMI, the risk of future diabetes. Moreover, intervention studies to improve postpartum eating practices in women with a history of GDM are highly warranted.

## Figures and Tables

**Figure 1 nutrients-15-04258-f001:**
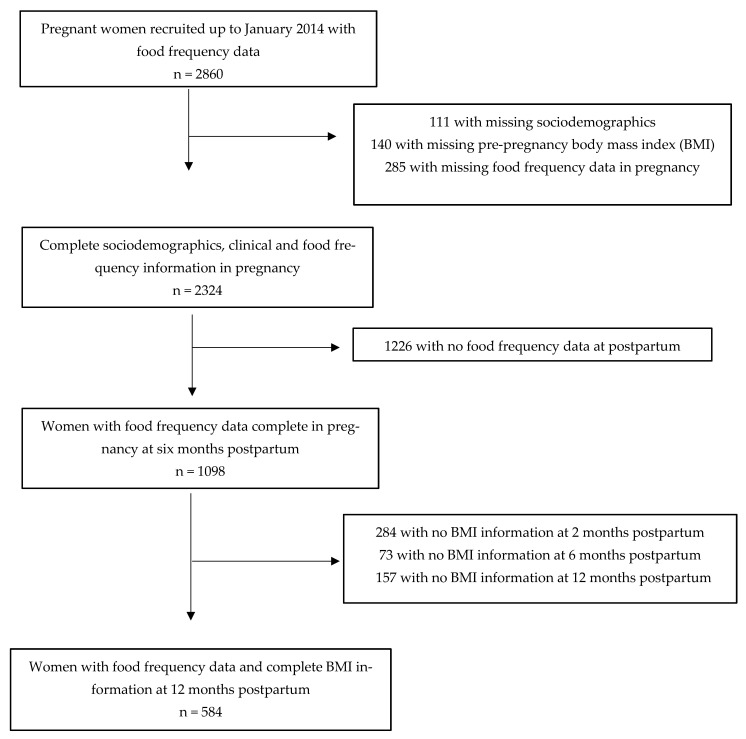
Sample size diagram.

**Figure 2 nutrients-15-04258-f002:**
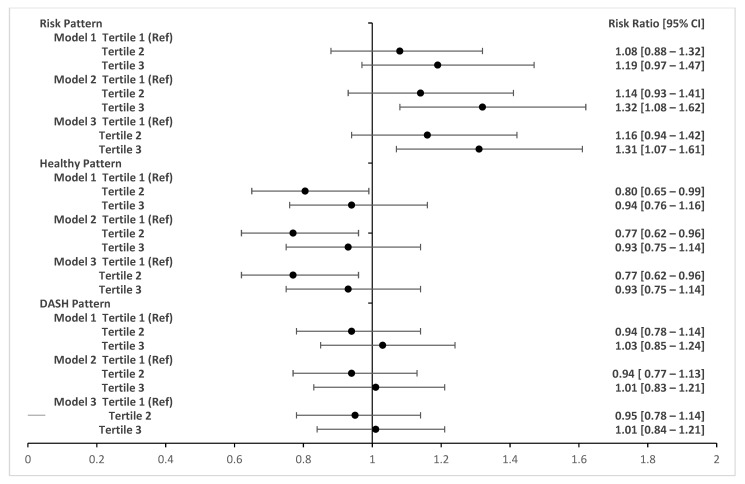
Adjusted associations between food patterns in pregnancy and BMI variation at two months and six months postpartum. LINDA-Brasil Cohort. RR (95% CIs) adjusted by Poisson regression with robust variance. Model 1: adjusted for age, skin color, family income, study center, smoking during pregnancy, and parity. Model 2: adjusted for model 1 plus pre-pregnancy BMI and gestational weight gain. Model 3: adjusted for model 2 plus breastfeeding during the follow-up period.

**Figure 3 nutrients-15-04258-f003:**
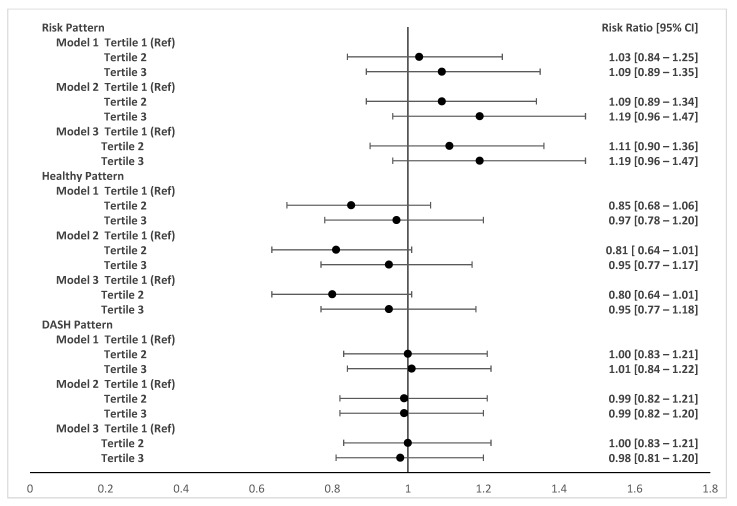
Adjusted associations between food patterns in pregnancy and BMI variation at two months and one year postpartum. LINDA-Brasil Cohort. RR (95% CIs) adjusted by Poisson regression with robust variance. Model 1: adjusted for age, skin color, family income, study center, smoking during pregnancy, and parity. Model 2: adjusted for model 1 plus pre-pregnancy BMI and gestational weight gain. Model 3: adjusted for model 2 plus breastfeeding during follow-up period.

**Figure 4 nutrients-15-04258-f004:**
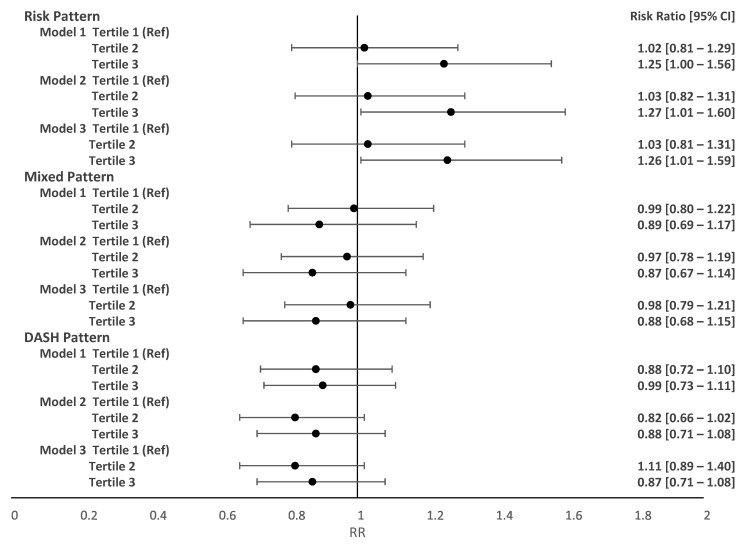
Adjusted associations between food patterns in the postpartum period and BMI variation at two months and six months postpartum. LINDA-Brasil Cohort. RR (95% CIs) adjusted by Poisson regression with robust variance. Model 1: adjusted for age, skin color, family income, study center, smoking during pregnancy, and parity. Model 2: adjusted for model 1 plus pre-pregnancy BMI and gestational weight gain. Model 3: adjusted for model 2 plus breastfeeding during the follow-up period.

**Figure 5 nutrients-15-04258-f005:**
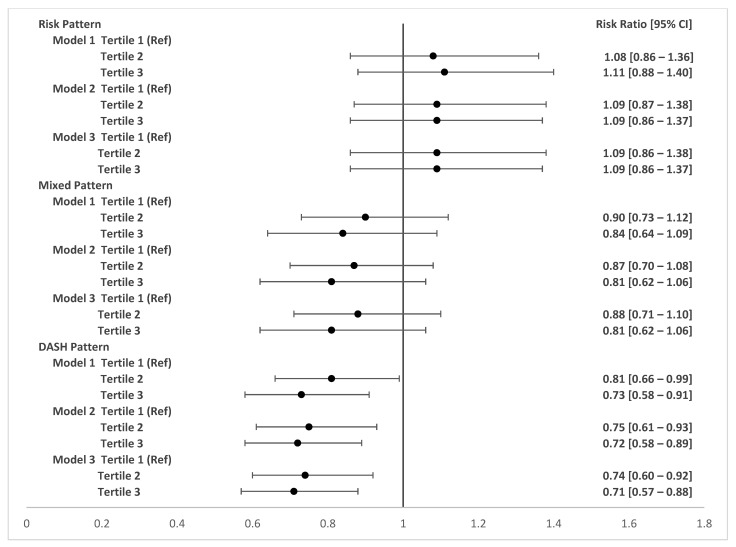
Adjusted associations between food patterns in the postpartum period and BMI variation at two months and one year postpartum. LINDA-Brasil Cohort. RR (95% CIs) adjusted by Poisson regression with robust variance. Model 1: adjusted for age, skin color, family income, study center, smoking during pregnancy, and parity. Model 2: adjusted for model 1 plus pre-pregnancy BMI and gestational weight gain. Model 3: adjusted for model 2 plus breastfeeding during the follow-up period.

**Table 1 nutrients-15-04258-t001:** Sample characteristics of women with complete information during pregnancy and postpartum. LINDA-Brasil cohort.

Sample Characteristics	Initial Sample n = 2324	With Complete Food Frequency Data n = 1098	With Weight Information at Follow-Up n = 584
	n (%)	n (%)	n (%)
Age at enrollment (years)			
18–29	791 (34.0)	395 (36.0)	202 (34.6)
30–39	1256 (54.0)	586 (53.4)	323 (55.3)
40+	277 (11.9)	117 (10.7)	59 (10.1)
Self-declared skin color			
White	1392 (59.9)	594 (54.2)	295 (50.8)
Non-white	932 (40.1)	501 (45.8)	286 (49.2)
Living with partner			
No	255 (11.0)	102 (9.3)	45 (7.7)
Yes	2069 (89.0)	996 (90.7)	539 (92.3)
Family income (minimum wage)			
<1	495 (21.3)	207 (18.9)	90 (15.4)
1 to <2	908 (39.1)	442 (40.3)	218 (37.3)
2 to <3	499 (21.5)	244 (22.2)	144 (24.7)
≥3	422 (18.2)	205 (18.7)	132 (22.6)
Parity			
0	527 (22.7)	241 (21.9)	138 (23.6)
1–2	1201 (51.7)	573 (52.2)	308 (52.7)
≥3	596 (25.6)	284 (25.9)	138 (23.6)
Self-reported smoking before pregnancy			
No	1836 (79.0)	849 (77.3)	446 (76.4)
Yes	487 (21.0)	249 (22.7)	138 (23.6)
Pre-pregnancy BMI (kg/m^2^)			
Normal and underweight (<25)	380 (16.4)	200 (18.3)	111 (19.1)
Overweight (25 to <30)	890 (38.4)	417 (38.1)	228 (39.2)
Obesity (≥30)	1048 (45.2)	477 (43.6)	243 (41.8)
Gestational weight gain (IOM 2009)			
Insufficient	1149 (54.4)	556 (52.1)	305 (53.2)
Adequate	526 (24.9)	262 (24.9)	145 (25.3)
Excessive	439 (20.8)	235 (22.3)	123 (21.5)
BMI at 2 months postpartum (kg/m^2^)			
Normal and underweight (<25)	203 (12.8)	108 (13.3)	83 (14.2)
Overweigh (25 to <30)	633 (40.0)	349 (42.9)	257 (44.0)
Obesity (≥30)	745 (47.1)	357 (43.9)	244 (41.8)
Breastfeeding at 6 months			
No	378 (34.8)	378 (34.7)	191 (32.8)
Yes	709 (65.2)	710 (65.3)	391 (67.2)
Breastfeeding at 12 months			
No	592 (47.6)	420 (47.0)	279 (48.1)
Yes	652 (52.4)	474 (53.0)	301 (51.9)

BMI = body mass index, IOM = Institute of Medicine, 2009. Gestational Weight Gain Recommendation.

**Table 2 nutrients-15-04258-t002:** Food frequency consumption during pregnancy and at six months postpartum. LINDA-Brasil cohort.

Food Items	Initial Sample n = 2324			With Complete Food Frequency Data n = 1098			
	Pregnancy			Postpartum (6 Months)			
	0–2 Days a Week	3–4 Days a Week	5–7 Days a Week	0–2 Days a Week	3–4 Days a Week	5–7 Days a Week	*p*-Value *
Vegetables, fruits and beans							
Raw salad	509 (21.9)	711 (30.6)	1104 (47.5)	242 (22.0)	439 (40.0)	417 (38.0)	<0.001
Cooked salad	762 (32.8)	1055 (45.4)	507 (21.8)	341 (31.1)	568 (51.8)	188 (17.1)	<0.001
Fruits	188 (8.1)	489 (21.0)	1647 (70.9)	249 (22.7)	362 (33.0)	487 (44.4)	<0.001
Natural fruit juice	1082 (46.6)	581 (25.0)	660 (28.4)	683 (62.3)	211 (19.2)	203 (18.5)	<0.001
Beans	177 (7.6)	476 (20.5)	1671 (71.9)	89 (8.1)	258 (23.5)	751 (68.4)	<0.001
Dairy products							
Low-fat dairy	1377 (59.3)	196 (8.4)	751 (32.3)	845 (83.8)	143 (14.2)	20 (2.0)	<0.001
Full-fat dairy	1078 (46.4)	353 (15.2)	893 (38.4)	529 (56.8)	366 (39.3)	37 (4.0)	<0.001
Meat							
Red meat with visible fat	1738 (74.8)	460 (19.8)	125 (5.4)	776 (70.7)	264 (24.1)	57 (5.2)	<0.001
Red meat without visible fat	742 (31.9)	1193 (51.3)	389 (16.7)	350 (32.0)	588 (53.7)	157 (14.3)	<0.001
Chicken	292 (12.6)	1468 (63.2)	563 (24.2)	98 (8.9)	715 (65.3)	282 (25.8)	<0.001
Fish	1882 (81.0)	412 (17.7)	30 (1.3)	958 (87.2)	134 (12.2)	6 (0.5)	<0.001
High caloric and ultra-processed foods							
Fried foods	1815 (78.1)	393 (16.9)	115 (5.0)	889 (81.0)	190 (17.3)	19 (1.7)	<0.001
Salty snacks	1078 (46.4)	793 (34.1)	453 (19.5)	561 (51.1)	386 (35.2)	150 (13.7)	<0.001
Cookies and sweets	1600 (68.8)	456 (19.6)	126 (11.5)	713 (64.9)	297 (27.0)	88 (8.0)	<0.001
Sweetened beverages	1226 (52.8)	502 (21.6)	595 (25.6)	439 (40.0)	274 (25.0)	385 (35.1)	<0.001
Processed meat	1436 (61.8)	638 (27.5)	250 (10.8)	666 (60.7)	317 (28.9)	115 (10.5)	<0.001

* Pearson chi-square test.

**Table 3 nutrients-15-04258-t003:** Food patterns identified through principal component analysis during pregnancy and at six months postpartum. LINDA-Brasil cohort.

Food Patterns	Food Items	Factor Loading	% of Total Variance
Pregnancy			
Risk			15.938
	Fried foods	0.712	
	Cookies and sweets	0.683	
	Sweetened beverages	0.671	
	Processed meat	0.604	
	Red meat with visible fat	0.416	
Healthy			10.811
	Red meat without visible fat	0.680	
	Raw salad	0.635	
	Cooked salad	0.541	
	Fruits	0.272	
DASH			10.492
	Low-fat dairy	0.826	
	Raw salad	0.193	
	Cooked salad	0.173	
Postpartum			
Risk			12.103
	Fried foods	0.685	
	Cookies and sweets	0.613	
	Processed meat	0.521	
	Red meat with visible fat	0.510	
	Sweetened beverages	0.447	
Mixed			11.148
	Natural fruit juice	0.640	
	Salty snacks	0.461	
	Fruits	0.422	
	Chicken	0.338	
DASH			10.752
	Cooked salad	0.679	
	Raw salad	0.667	
	Low-fat dairy	0.503	
	Fruits	0.452	

Pregnancy: KMO = 0.686—Bartlett test of sphericity *p* < 0.001. Postpartum: KMO = 0.616—Bartlett test of sphericity *p* < 0.001.

## Data Availability

The data analyzed in this study are subject to the following licenses/restrictions: no restriction. Requests to access these datasets should be directed to midrehmer@hcpa.edu.br.

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
