# Peer review of "Dietary Patterns in Pregnancy and the Postpartum Period and the Relationship with Maternal Weight up to One Year after Pregnancy Complicated by Gestational Diabetes"

_nutrients, 2023, doi:10.3390/nu15194258_

Round 1
Reviewer 1 Report
The aim of the present study is to describe how the diet pattern changes between gestation and postpartum in woman with gestational diabetes and the association of these changes with BMI variation at six months and one year postpartum. The authors conclude that this cohort of women consumed more ultra-processed foods in the postpartum period than during gestation, and less salads, fruits, vegetables and diary products. In addition they conclude that healthy dietary pattern is associated with less weigh gain while risk dietary pattern is associated with higher weight gain. It is an interesting study, however, there are some aspects that need to be clarified.
Results
-In the results section, the authors state that frequent consumption of vegetables, fruits and legumes decreases in the postpartum period . However, the data show that only fruit consumption shows an important decrease. Tha authors should include the statistical significance in Table 2.
-The authors mention that adherence to risk dietary pattern during pregnancy and postpartum was associated with a 31% and 26% increase, respectively, in BMI between two and six monts postpartum. However, according to the authors, the diet worsens in the postpartum period (there is a higher frequency of consumption of sugary drinks in the postpartum period). How could it be explained that a higher frequency of consumption of sugary drinks in the postpartum period is associated with similar, or even lower, weight gain than that observed with the risk diet model during gestation?
-The authors write that “Examining individual food items, after adjustment, high intake of fried foods during pregnancy increased by 83%....”. Increased should be changed by “was associated” because the authors do not demonstrate that fried foods increases BMI. This should be revised throughout the paragraph.
-High fried food intake during pregnancy and during postpartum period is associated with the same increase in BMI at six months and 1 year postpartum. How would the authors explain this considering that high fried food intake decreased significantly during the postpartum period?. Similarly, how would the authors explain that high intake of sugar-sweetened beverages during pregnancy and during postpartum period is associated with a similar increase in BMI considering that the intake of this type of beverages is higher in the postpartum period.
Discussion
-The authors associate the type of diet during gestation with the increase in BMI in the postpartum period. The authors should discuss how diet during gestation influences BMI during the first year after delivery. Have the authors analyzed whether women who take a type of diet during gestation maintain it after delivery?. This would explain the association of diet type during gestation with the increase in BMI postpartum.
-In the discussion the authors conclude that during the postpartum period women increase the consumption of ultra-processed food, but the data show that they only increase the consumption of sugar-sweetened beverages. This should be corrected in the text.
Author Response
Dear Ms Yoki He
Section Managing Editor, Nutrients
Thank you for the thoughtful review, we believe it helped improve the manuscript. We have responded to reviewers’ comments and revised the manuscript, accordingly, indicating in red the changes we made. We also revised the text for possible misspellings and to improve clarity. The manuscript has undergone English language editing by MDPI. The text has been checked for correct use of grammar and common technical terms, and edited to a level suitable for reporting research in a scholarly journal.
Reviewer 1
Comments and Suggestions for Authors
The aim of the present study is to describe how the diet pattern changes between gestation and postpartum in woman with gestational diabetes and the association of these changes with BMI variation at six months and one year postpartum. The authors conclude that this cohort of women consumed more ultra-processed foods in the postpartum period than during gestation, and less salads, fruits, vegetables and diary products. In addition they conclude that healthy dietary pattern is associated with less weigh gain while risk dietary pattern is associated with higher weight gain. It is an interesting study, however, there are some aspects that need to be clarified.
Results
-In the results section, the authors state that frequent consumption of vegetables, fruits and legumes decreases in the postpartum period . However, the data show that only fruit consumption shows an important decrease. Tha authors should include the statistical significance in Table 2.
Response: Thank you, we added P-values in Table 2 according to your suggestion.
-The authors mention that adherence to risk dietary pattern during pregnancy and postpartum was associated with a 31% and 26% increase, respectively, in BMI between two and six monts postpartum. However, according to the authors, the diet worsens in the postpartum period (there is a higher frequency of consumption of sugary drinks in the postpartum period). How could it be explained that a higher frequency of consumption of sugary drinks in the postpartum period is associated with similar, or even lower, weight gain than that observed with the risk diet model during gestation?
Response: Thank you for this comment. Dietary patterns were derived for each period, thus we are assessing associations for dietary patterns derived from food consumption during pregnancy and the postpartum period. Although we referred them as risk dietary pattern, we ran two different factorial analyses and the high factor loadings of the foods were different (see Table 3). In pregnancy, the food items and respective factor loadings were fried foods 0.712, cookies and sweets 0.683, and sweetened beverages 0.671. In postpartum, the food items and respective factor loadings were fried foods 0.685, cookies and sweets 0.613 and processed meat 0.521. Thus, relative risks between these patterns and the outcome also differed.
-The authors write that “Examining individual food items, after adjustment, high intake of fried foods during pregnancy increased by 83%....”. Increased should be changed by “was associated” because the authors do not demonstrate that fried foods increases BMI. This should be revised throughout the paragraph.
Response: Thank you for this suggestion. We made the correction.
-High fried food intake during pregnancy and during postpartum period is associated with the same increase in BMI at six months and 1 year postpartum. How would the authors explain this considering that high fried food intake decreased significantly during the postpartum period?. Similarly, how would the authors explain that high intake of sugar-sweetened beverages during pregnancy and during postpartum period is associated with a similar increase in BMI considering that the intake of this type of beverages is higher in the postpartum period.
Response: Good point. When we examine the relationship of only one food item rather than looking at a dietary pattern, we fail to assess the overall diet. Therefore, although there was a decrease in the consumption of fried foods, factors other than just fried foods may be contributing to weight gain (there are certainly residual confounders) in the postpartum period. Likewise, although there was a higher consumption of sweetened drinks in the postpartum period compared to pregnancy, we found similar measures of association with weight gain in the postpartum period among pregnant women who consumed sugar-sweetened beverages, because this eating behavior may be also associated with other factors of risk for weight gain than just sugar-sweetened beverages.
Discussion
-The authors associate the type of diet during gestation with the increase in BMI in the postpartum period. The authors should discuss how diet during gestation influences BMI during the first year after delivery.
Response: We discussed how diet during gestation influences BMI during the first year after delivery in the fourth paragraph of the discussion, as follows:
“Evidence points out that adherence to healthy dietary patterns during pregnancy, such as the Mediterranean diet, rich in antioxidants and phenolic compounds, the New Nordic Diet, based on consumption of fruits, root vegetables, cabbages, potatoes, oatmeal, whole grains, wild fish, berries and milk and Dash Diet, that is rich in fruits, vegetables, whole grains, and low-fat dairy products, protects against postpartum weight gain (30,31,15)”. Most studies do not show a long postpartum follow-up period (one year or more, for example). However, two important studies bring information about healthy eating during pregnancy and less weight gain up to 8 years post-delivery (31) and another retrospective study pointed out that diet change (most improvement in quality) gained significantly less weight per 4-year period (15).
-Have the authors analyzed whether women who take a type of diet during gestation maintain it after delivery?. This would explain the association of diet type during gestation with the increase in BMI postpartum.
Response: Regarding food consumption evaluation in our study, we only questioned the food items present in the FFQ. We did not question whether the pregnant woman was following any dietary plan prescribed during prenatal care. Although, we know that this population of pregnant women enrolled in high-risk prenatal services receive nutritional and endocrinological follow-up, many of them must follow a prescribed diet.
-In the discussion the authors conclude that during the postpartum period women increase the consumption of ultra-processed food, but the data show that they only increase the consumption of sugar-sweetened beverages. This should be corrected in the text.
Response: We appreciate the comment and changed the manuscript accordingly as:
“We also found a high-risk dietary pattern based on high-calorie foods and sweetened drinks may increase risk of postpartum weight gain”.

Reviewer 2 Report
This is an interesting paper to read, with sound methods and results. There are many typographical errors / poor English quality that limits understanding.
There are no P-values reported in the analysis results, is it ok to assume the research can interpret the level of significance from the 95%CI's?
The conclusion does not say in what why the results contribute to the literature in this area, can the authors provide more information about your findings. For example, the worsening of the dietary patterns postpartum is an important finding.
The authors need to review their tables, there are many formatting errors / inconsistencies.
In the introduction, there is a typographical error - " associated with the lowest gain weight loss" do you mean the lowest gain or the lowest weight loss?
In the conclusion this sentence is not clear " More prospective studies that assess, in addition to BMI, also the risk of future diabetes, are needed."
An error in table 5 the words "Tertile 2" are misaligned
Is reference 21 correctly listed?
Reference 31 journal has not been abbreviated. Authors to check reference list
Author Response
Dear Ms Yoki He
Section Managing Editor, Nutrients
Thank you for the thoughtful review, we believe it helped improve the manuscript. We have responded to reviewers’ comments and revised the manuscript, accordingly, indicating in red the changes we made. We also revised the text for possible misspellings and to improve clarity. The manuscript has undergone English language editing by MDPI. The text has been checked for correct use of grammar and common technical terms, and edited to a level suitable for reporting research in a scholarly journal.
Reviewer 2
Comments and Suggestions for Authors
This is an interesting paper to read, with sound methods and results. There are many typographical errors / poor English quality that limits understanding.
Response: We appreciate the comment, and we revised the text throughout for possible misspellings and to improve clarity.
There are no P-values reported in the analysis results, is it ok to assume the research can interpret the level of significance from the 95%CI's?
Response: Thank you for your comment. We have added in methods the following sentence: “We presented risk ratios followed by 95% confidence interval (95% CI). This allows assessment of precision and indicates the lower and upper bounds that may fall within the statistical significance range.
The conclusion does not say in what why the results contribute to the literature in this area, can the authors provide more information about your findings. For example, the worsening of the dietary patterns postpartum is an important finding.
Response: Thank you for your suggestion. We have included in the conclusion section this phrase: “Worsening the dietary patterns from pregnancy to postpartum results in postpartum maternal weight gain, which has a greater chance of enhancing the risk of T2DM in women with a history of GDM”.
The authors need to review their tables, there are many formatting errors / inconsistencies.
Response: We appreciate the comment and we have reviewed all tables carefully for improvement.
Comments on the Quality of English Language
In the introduction, there is a typographical error - " associated with the lowest gain weight loss" do you mean the lowest gain or the lowest weight loss?
Response: Thank you for your comment. We made the correction in the introduction.
“…was significantly associated with less weight gained…”
In the conclusion this sentence is not clear " More prospective studies that assess, in addition to BMI, also the risk of future diabetes, are needed."
Response: Thank you for your comment. We made the correction in the conclusion.
“Further prospective studies are needed to assess, in addition to BMI, the risk of future diabetes".
An error in table 5 the words "Tertile 2" are misaligned
Response: We made the correction.
Is reference 21 correctly listed?
Response: Yes, it is correct.
Reference 31 journal has not been abbreviated. Authors to check reference list
Response: Thank you. We made the correction and checked all reference list.
